# Automated Detection of Multi-Rotor UAVs Using a Machine-Learning Approach

**Šimon Grác, Peter Beňo, František Duchoň \***[ID]**, Martin Dekan and Michal Tölgyessy**

Institute of Robotics and Cybernetics, Slovak University of Technology, 812 19 Bratislava, Slovakia;
samgrac@gmail.com (Š.G.); beno@photoneo.com (P.B.); martin.dekan@stuba.sk (M.D.);
michal.tolgyessy@stuba.sk (M.T.)
**\*** Correspondence: frantisek.duchon@stuba.sk

**Abstract:** The objective of this article is to propose and verify a reliable detection mechanism of multi-rotor unmanned aerial vehicles (UAVs). Such a task needs to be solved in many areas such as in the protection of vulnerable buildings or in the protection of privacy. Our system was firstly realized by standard computer vision methods using the Oriented FAST and Rotated BRIEF (ORB) feature detector. Due to the low success rate achieved in real-world conditions, the machine-learning approach was used as an alternative detection mechanism. The "Common Objects in Context dataset" was used as a predefined dataset and it was extended by 1000 samples of UAVs from the SafeShore dataset. The effectiveness and the reliability of our system are proven by four basic experiments—drone in a static image and videos which are displaying a drone in the sky, multiple drones in one image, and a drone with another flying object in the sky. The successful detection rate achieved was 97.3% in optimal conditions.

**Keywords:** UAV; detection; machine learning; TensorFlow; ORB

---

## 1. Introduction

Object recognition algorithms and their classification are based on the fact that the considered objects have common characteristics. This means that features are not defined only by their appearance but also by the behavior or the way of movement. The issue of unmanned aerial vehicle (UAV) recognition in the sky is focused on objects that could occur in the sensing area as well as on the UAVs themselves. One of the basic visual properties of a UAV is its shape. Each type of UAV (from the tricopter to the octocopter) looks very similar in its category. A tricopter has the shape of an equilateral triangle, a quadcopter has the shape of a square, etc. Moreover, each UAV is composed of rigid construction that has its own visual characteristics. This construction usually includes storage space for the control board in the middle and from three to eight arms depending on the number of propellers. Each arm is finished by the motor on which the propeller is suspended. This is the basic UAV appearance pattern which can be used for almost every single UAV. An exception may be the UAVs from the nano and mini categories, where the propeller and the motors can be mounted directly on the center panel of the referenced UAV.

One of the main advantages of these machines is their extended battery life which makes this drone type able to operate for a longer time in the air [1] and also fly higher [2]. Some of the fixed-winged drones as well as the multi-rotor models can take off from the ground, but most of them can take off and stay in the air only during movement.

There is also another type of drone that differs from the multi-rotor models in size, weight, and the maximum achievable speed. These drones are called fixed-winged drones [3]. Due to their properties, they are often used in applications such as environment and area mapping (with the possibility of

further processing and generation of three-dimensional data [4]), in meteorology [2] or for inspection of quality (an interesting example is the inspection of electrical wiring described in [5]).

According to [1], fixed-winged UAVs are suitable for the mentioned applications mainly because of their purchase price, maintenance, and operational time. According to [2], another significant advantage is their operation range and their flight safety. Using multi-rotor models, according to [1], there is a better possibility of specific environmental mapping and the main advantages include better use in civilian applications and simple maneuvering. As stated in [6], the growing trend of using these machines will require a certain amount of research and regulation in the future, to ensure their safe usage without restricting airspace, traffic flows, and their efficient management.

UAVs are also characterized by the presence (or absence) of several individual components. Some UAVs, for example, are covered by a ring located around the entire periphery of the machine. On the other hand, other UAV types have some parts completely uncovered. Some types of UAV also use an additional covering ring, which may or may not be located in the vicinity of the machine propellers. Most UAVs are equipped with a camera. The camera can be located within the body of the UAV, where it becomes visually less noticeable. However, on UAVs with better-quality cameras, the camera is located on the bottom of the UAV, where it forms a large visually noticeable part. Several types of UAV are characterized also by additional devices. For example, in large-scale UAVs, there are often accessories such as robotic arms or other devices. These aspects, i.e., large additional devices, are not considered in the proposed detection and identification procedure.

The main purpose of this work is to design and implement an automated detection approach whose properties and reliability can be compared with the common human observer staring at the sky. The proposed approach will be able to recognize multi-rotor drones based on their appearance. We focus our attention on these drone models mainly because of their versatility of use in common civil applications.

Considering the detection of the UAV itself, the fact that some UAVs have a light device on the lower part of the body can make the detection more effective especially during unfavorable illumination conditions. Another feature that characterizes a UAV is the way of movement. The purpose of the movement is to move the UAV from the stabilized position from the start point to the endpoint at which the machine is to be stabilized again. The UAV movement itself is characterized by its monotony. Contrary to the movement of a bird sliding or waving its wings, the UAV does not perform any such movements.

The basic analysis of a UAV's movement was performed in our research, and from the results it is clear that these machines are moving according to certain patterns. The UAV movement is mostly linear but it can suddenly change direction, speed, or height. The UAV can also stabilize its position and float in place without moving. These movements are considerably different from those of birds.

There have been several reports of UAV abuse for privacy or terrorist attacks. Some cases are also known where a UAV has been used for bringing contraband materials into prisons. These, as well as many other potential misuses of these machines, create the need to identify and track UAVs in an area where such attacks may occur. Due to the features and versatility of the use of these machines, there are many possibilities for their misuse. These machines can fly very low to the ground, but also very high. They can disguise themselves behind various objects and with the help of additional devices, they can fly in different lighting conditions and overcome various obstacles. However, we assume that these machines would have a problem operating during adverse conditions such as high wind, rain, snow, smog, or hail. We do not deny the possibility of using machines that are adapted to such conditions, but in our work, we assume a significantly reduced ability to maneuver or use additional equipment.

Because the UAVs can move quickly and undetected in many areas they can create significant risks. The UAV thus becomes an ideal machine for disrupting safety, endangering life, causing damage, or making an unauthorized entry to private land or tracking people. For these reasons, different regulations arise. In the Slovak Republic Decision no. 2/2019 from 14 November 2019 refers to the

precise division of UAVs into groups to which specific regulations apply. It also refers to the precisely defined conditions under which unmanned aircraft can be used in the airspace of the country.

In general, we can divide the basic approaches of UAV detection into visual and acoustic. Visual detection is realized by camera systems capable of recording a two-dimensional image (cameras operating in the dark can be used as well). These approaches could be based on the appearance or movement of the object, or their combination. Acoustic methods are using microphones [7]. An interesting advantage of acoustic methods is the possibility of detection even when the drone is not in sight [8]. These two approaches can be also combined to create a hybrid detection system. An example is an anti-drone system described in [9], which uses a combination of audio, video, radio-frequency sensors, and a radio-frequency jamming unit.

The article is structured as follows: The second section introduces the existing methods for object recognition in an image. The basic analysis of several methods is described. The third section describes the proposed drone detection system that uses standard computer vision methods. The fourth section describes another detection mechanism that uses TensorFlow. The fifth section introduces several experiments with the evaluation of the success rate. The sixth section concludes the paper and proposes future work.

## 2. Existing Methods for Detection and Identification of Objects in an Image

### 2.1. Background Subtraction

For ease of operation in particular, the background subtraction method is one of the most basic methods for detecting objects in an image. As described in [10], this method needs to accurately identify the background model. After completing this step, the background model is compared to the current image, and the known background parts are subtracted. Objects that are not subtracted are with certain probability new objects in the foreground. Normally the background is defined as any static or periodically moving parts of the scene. The entire scene may have time-varying components, such as tree leaves that move at some time but are static at another time. A common element of systems whose purpose is to monitor the objects by a static camera is a module whose task is to subtract a background to distinguish static objects from dynamic objects. A substantial and complex part of the background subtraction process is maintaining the background model. According to [11], there are some situations when it is difficult to read or detect the background: inhomogeneous and variable illumination of the scene, changing spectral characteristics of the illumination and consequently different color of the object, overlapping objects, different camera angles, and object variations within one category. In [12] a detailed comparison of different background subtraction techniques is elaborated. The purpose of this comparison was to find out which of the techniques could cope best with the problems mentioned above.

### 2.2. Contour Searching

The basic idea of using contours in image processing is to produce a curve that encloses the objects contained in the image. Successful usage of this object bounding method is dependent on applied image preprocessing methods such as image smoothing and morphological operations. The only condition for using contour searching is to divide the image into so-called positive and negative regions [10], whose boundaries can be considered as bounded objects. The correct setting of the parameters in this method guarantees the correct detection of contours in the image. The term contour is defined as a list of points that represent a curve in an image. These curves are represented as sequences in which the record encodes information about the next point on the curve. Due to its structure, the contour searching function can construct a so-called contour tree. Therefore, it can determine which contour is a root contour and which contours represent the child contours [10,13]. In the case of detection of objects in the image, in most cases, it is necessary to delimit only the root contour of the object.

In Figure 1 the segmented object is shown on the left and the object whose contours are outlined is shown on the right. It is also possible to see several other contours of different colors that represent other objects inside of the root contour. This phenomenon can also occur after morphological operations and image smoothing since the recorded objects are never homogeneous.

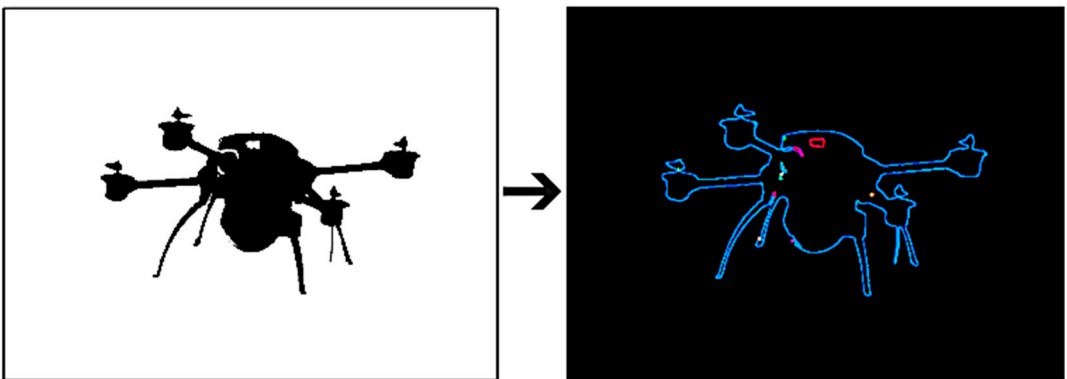

**Figure 1.** Contour searching applied to the edited image with a drone.

### 2.3. Selective Searching

One of the most effective ways to find subregions containing an object in an image is an algorithm called selective searching. As reported in [14], this algorithm is based on three main assumptions:

1.　Capturing all possible scales in the image—using a hierarchical algorithm, selective searching attempts to take into account all possible scales of the objects;
2.　Diversification—since objects in the analyzed area are subject to different changes such as illumination, shadows, and other, selective searching does not use a uniform strategy for a subregion search;
3.　Calculation speed—since the step of subregion searching is only a preparation for the object recognition itself, this algorithm is designed to not cause any decrease of calculation speed.

An example of subregion searching in a static image using a selective searching algorithm is shown in Figure 2.

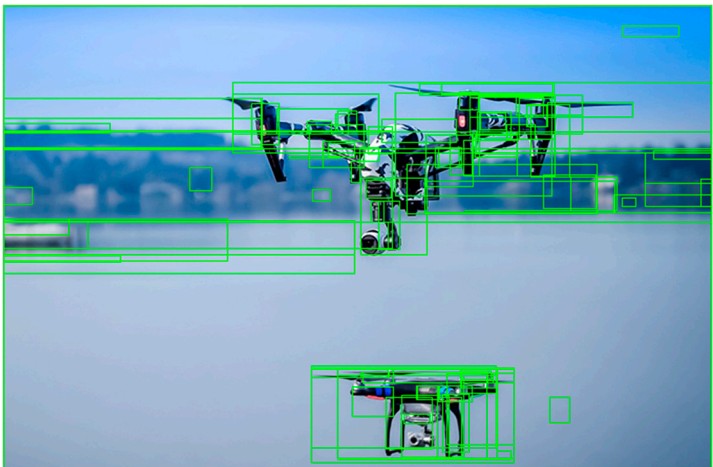

**Figure 2.** Selective searching in the image with drones.

### 2.4. Support Vector Machines (SVM)

As mentioned in [10], support vector machines (SVMs) are suitable for assigning objects to N groups and their functionality is based on projecting data into multidimensional space. SVM search

for and determine the plane through which it splits data into groups. For example, if there is a vector of features with dimension 2500, SVM would represent this vector as a point in a space with 2500 dimensions [15]. For the sake of simplicity, let's imagine the SVM functionality for the vector of features with 2 dimensions. The visualization of SVM decomposition, in this case, can be seen in Figure 3.

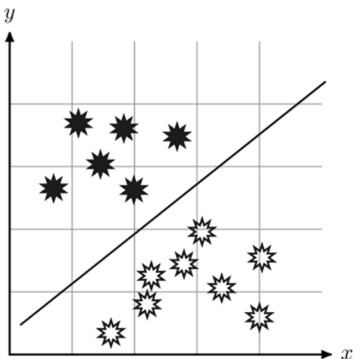

**Figure 3.** Support vector machine (SVM) principle.

As can be seen in Figure 3, the line represents a classifier (for example a solid shape can represent the drone and shape with fill represents the background) and it divides the data successfully into two classes. Sometimes, when classifying with similar methods, the problem is that the superposition splitting data is too close to one of the classes. Therefore, SVM seeks a separator to reach the maximum distance between the data classes. If the classes are non-separable, the so-called non-linear SVM is used. In this case, the data are projected into a multidimensional space where they are separable [10].

### 2.5. Cascade Classifier (Haar-Like Features)

This classifier is primarily intended for the classification of objects that are stable [10]. As an example, the figure of the human body or human face can be mentioned. The human body and face in most cases have the same proportions, i.e., there are always hands, legs, head, and so on. The drone shape analysis showed that the drone does not fall into such a category of objects. As stated in [16], there are two main reasons why a Haar classifier is used to recognize objects. One is that Haar-like features can effectively describe a region of interest, which is a challenging task with limited training data. If Haar features are compared to raw pixels, they can increase or reduce the variability of data belonging or not into a common class by their properties. Haar features are capable of recognizing and effectively describing the value of the ratio between light and dark parts of the scene. As mentioned in [16], they are also successful in classical computer vision problems, namely scene variability and varying illumination of the scene. The second reason to use this method of classification is its speed, as working with Haar features is considered highly effective.

### 2.6. Machine Learning and Neural Networks

The topic of using neural networks for object recognition is very complex. To use a neural network for this purpose, it is necessary to have a sufficiently large set of data showing the object that is needed to be recognized. Some of the methods using neural networks also require samples of data where the object to be recognized is not present. An error that the neural networks assign the object to a category is used to change the weights of the neurons so that the global error of the network gradually decreases during the network training. Such type of training is also called the error propagation algorithm. If the network reaches the specified error threshold, network training is terminated. Another possibility is to terminate the training after reaching a predetermined number of iterations. However, this solution does not take into account the global error of the training in any way.

*2.7. TensorFlow (TF)*

TensorFlow (TF) is an open-source machine learning platform that is used in a wide range of applications [17,18]. A tensor is a generalization of vectors and matrices to potentially higher dimensions. TF represents tensors as n-dimensional coordinates of basic data types [19]. This system is based on artificial intelligence and it was published by Google (for free use) in 2015. TF uses the dataflow graph method to represent the calculations. Units of calculations are represented by graph nodes. The edges of the graph transmit tensors (multidimensional arrays) between nodes [17,18] and represent the data consumed or produced by the calculation [20]. The advantage of this system is its architecture which by its flexibility allows easy and affordable deployment of computing on various platforms such as multicore processors, graphics cards, or Tensor processing units (a computer system also developed by Google designed primarily for machine learning). As mentioned in [17], training a neural network to classify and recognize objects requires a large number of calculations. The features of TF allow its users to perform computationally complex tasks such as classification in a relatively reasonable time.

## 3. Drone Detection Using Computer Vision Methods

Our detection procedure shall meet the following requirements and objectives: recognition of moving objects in the image, identification of each object in an image, correct matching of objects from the previous frame to the currently located objects in the image, and path drawing of the object in the scene. For these reasons, the detection procedure using computer vision was proposed as illustrated in Figure 4.

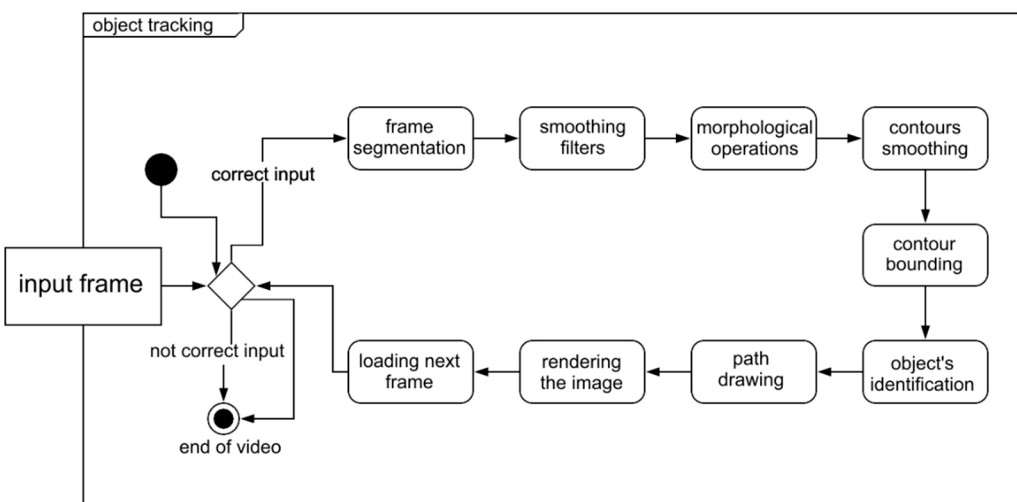

**Figure 4.** The basic scheme of the proposed detection procedure.

The first step is the background segmentation (Figure 5). Among the tested methods, the MoG (Mixture of Gaussians) method was chosen for the following reasons: adaptability of the algorithm to changes in illumination conditions during the day (not sudden changes as switching on the light in a room), a small movement of background objects that do not represent significant objects in the image and possible image covered by a large object. All of the cases can be expected during drone detection in the image.

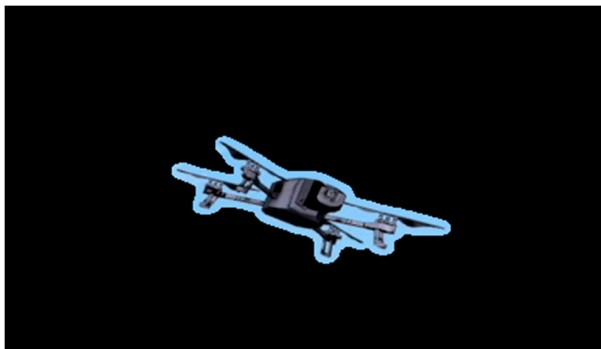

**Figure 5.** Segmentation of objects using the MoG (Mixture of Gaussians) method (background has black color).

The next step is the application of filters on the input image. It is needed to remove noise and smooth it out. Comparing the individual smoothing methods, a Gaussian filter, which [21] is considered to be "most useful", although not the fastest, was used due to the high noise reduction efficiency.

However, the correct selection of the image segmentation method and proper smoothing filter may still cause there to be areas in the image that do not match any of the objects. They simply represent noise caused by illumination conditions, camera focusing, or other reasons. The basic operations of mathematical morphology (erosion, dilatation, and morphological opening) were used to reduce the noise and correct bounding of the areas that belong to the objects in the image.

The contour search method determines whether or not the moving objects are in the image. A contour-bounding method using a rectangle was used to outline the moving object (Figure 6).

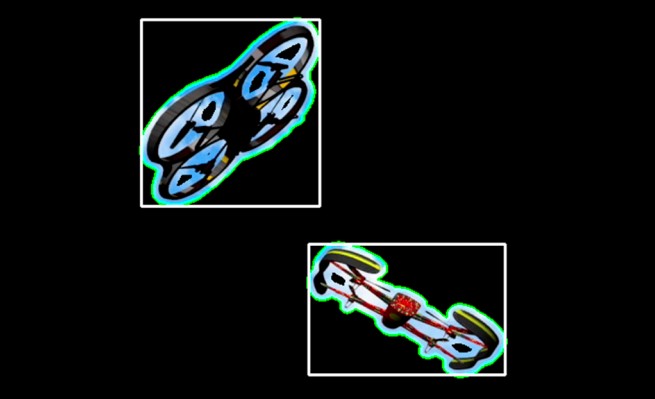

**Figure 6.** Moving objects bounded by the contour search method using a rectangle.

Successful tracking of moving objects requires the knowledge of whether the segmented object from the previous frame is the same as that detected in the current frame. If only one object is considered, this issue can be neglected. Of course, such a situation can occur, but it is necessary to take into account situations where there are many objects in the image. For this reason, the objects in the image must be identified, and to select the correct method, tracked objects must be defined. Knowing this information, an identifier can be defined. The identifier is the information that clearly describes the object being tracked. The tracked object must be trackable also in the presence of other objects even in the change of location, illumination conditions, or other changes. At this point, local features are appropriate to locate and describe areas in the image that belong to a particular object. The process of finding these features consists of the detection of such features and describing their surroundings. The methods by which these local features are searched to create descriptive vectors of features that are invariant (change in position, rotation, and other changes) are described in [11]. Several local feature detectors are available in the Open Computer Vision (OpenCV) library and have been tested and

analyzed. The following detectors were compared: Scale-Invariant Feature Transform (SIFT), Speed Up Robust Features (SURF), Binary Robust Invariant Scalable Keypoints (BRISK), Oriented FAST and Rotated BRIEF (ORB), and Accelerated KAZE (AKAZE). Two images showing one object (drone) were used for this analysis (Figure 7).

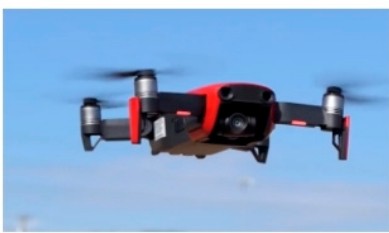 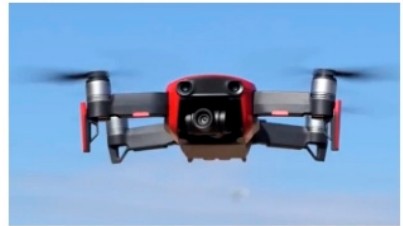

**Figure 7.** Testing images used to evaluate the suitability of feature detectors.

When testing each of the detectors, features were found in both frames, and then the vectors were extracted using the same descriptor, in this case, the SURF descriptor. Using the closest neighbor search method or algorithm Fast Library for Approximate Nearest Neighbors (FLANN), it was determined which feature in the first frame coincided with the feature identified in the second frame. Of these matches, those found to be good were selected. The good match was defined by this criterion: the mean distance between the feature identified in the first frame and feature identified in the second frame. Only those distances that were above the mean of maximum and minimum distances of all features were selected. Thus, only those that are qualitatively above half of all distances have been obtained. The result of this calculation was influenced by a suitably chosen constant. During the testing it was found that this method works relatively correctly and even though sometimes incorrectly identified features matches were found, it is considered to be a basic filter to determine the correct matching of features. Testing detectors led to the following findings:

1. *SIFT*—the number of features found compared to other detectors was higher, features were scattered throughout the object and identified also in areas that did not correspond to the edges of the object, finding good matches of features was also difficult after the application of additional filter;
2. *SURF*—as in the first detector, the number of features identified was too high and often did not correspond to the object's edges; features matching was only partially successful;
3. *BRISK*—the number of features found was higher, but most of them corresponded to the edges of the object and the important parts of the objects; a sufficient number of successfully matched features were achieved by the additional filter;
4. *ORB*—the number of significant points was the lowest among the tested detectors, but their localization was almost exclusively at the edges of the object and the important parts of the object; the points were not scattered throughout the whole object when the additional filter was applied; and a high number of correctly matched features was achieved;
5. *AKAZE*—the number of significant points was higher, some even outside the object and important areas; a sufficient number of successfully matched features were achieved by an additional filter.

The evaluation of detector testing is shown in Table 1. The greater the number of characters x, the more successful the test was.

**Table 1.** The evaluation of detectors test.

|  | SIFT | SURF | BRISK | ORB | AKAZE |
|---|---|---|---|---|---|
| Number of features | x | x | xx | xxx | xx |
| Dispersal of features | x | x | xxx | xxx | xx |
| The number of successfully matched features | x | x | xx | xx | xx |
| Overall rating | x | x | xx | xxx | xx |

Since some incorrect feature matchings were evaluated as good even after applying the good match filter, it would not be appropriate to compare the ratio of features found and well-matched features between the detectors. Thus, the success of each detector was compared visually. BRISK and ORB detectors achieved the best results. The ORB feature detector, which uses the BRIEF descriptor for feature extraction, was selected.

The goal of this work is not to compare existing detectors and evaluate their success in general. Therefore, we chose a representative pair of images describing our use case and tried to evaluate the success rate of detectors supported by the OpenCV library. A detailed description of used detectors, together with the achieved success, in general, is described in [22] in detail. Comparing our testing results and results published in [22], there is a required match in parameters that are important for us. Taking into account this information we consider our testing relevant and selected ORB detector suitable for our usage. The following test scenarios have been developed to verify the proposed procedures:

- movement and tracking of a single object,
- movement and tracking of multiple objects,
- leaving the sensing area,
- clash of objects.

During the first test, there was only one object in the test area that moved freely. The test also simulated a situation where the object once came out of the sensing area and returned at a different angle. Also, zooming in and out of objects was simulated, which would mean the object's approaching and moving away. Throughout testing, only one object in the test area was identified, whose identifier (the object itself) was updated in the database with each additional incoming frame. The result of the experiment can be seen in Figure 8.

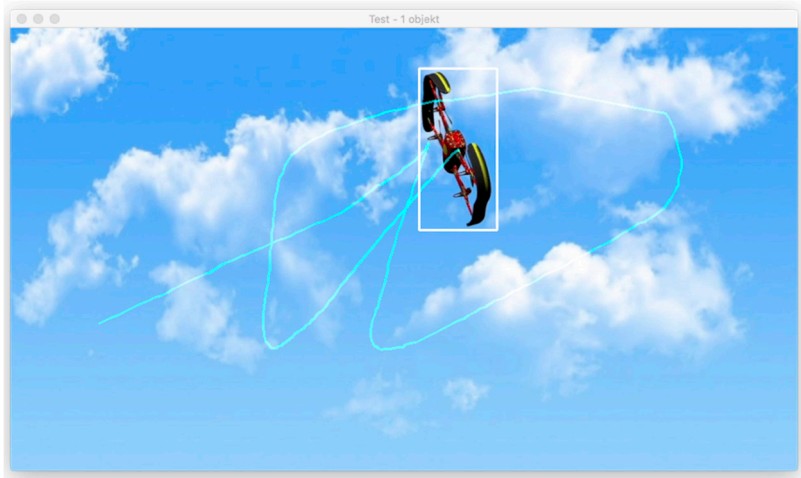

**Figure 8.** Movement and tracking of a single object using the ORB detector.

During further testing, two objects were in the sensing area. The objects, as in the first case, moved freely in the sensing area. The object described by the blue line (Figure 9) once left the area and returned. The proposed detection procedure was successful because the object (drone) was identified as the same. In Figure 9 it is clear that the object from the previous frame was always assigned to the correct object in the current frame.

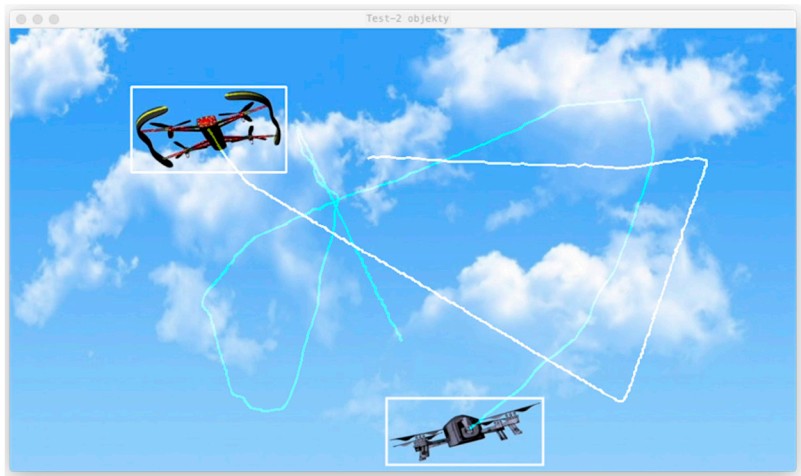

**Figure 9.** Movement and tracking of multiple objects using the ORB detector.

In the next scenario, the identified object left the sensing area. As shown in Figure 10, one object in the area has been identified. This object came out of the area several times and returned at a different angle, rotated, and did so at a different size (UAV was approaching and moving away from the camera). As is evident from the shape analysis, the drone is an invariant object and, in many cases, it resembles a different object. For this reason, the proposed solution is considered successful in this scenario.

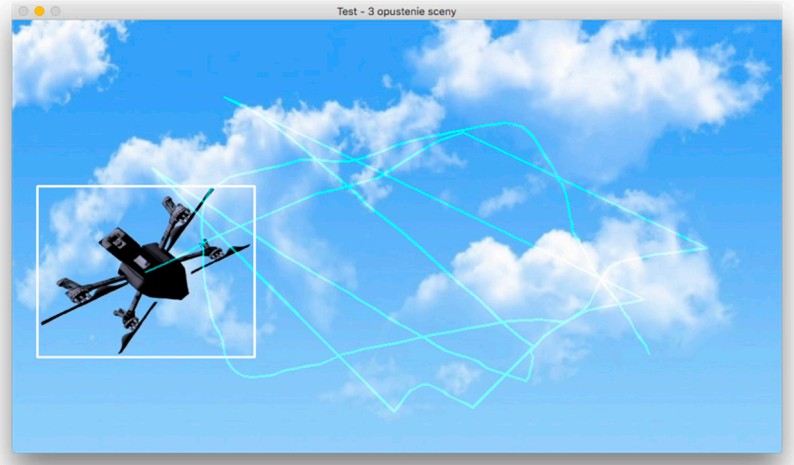

**Figure 10.** Leaving the sensing area.

When testing the objects clash scenario, several objects appeared on the scene that partially overlapped each other, and their path merged. All objects moved freely for a while and overlapped after some time. The test result is shown in Figure 11.

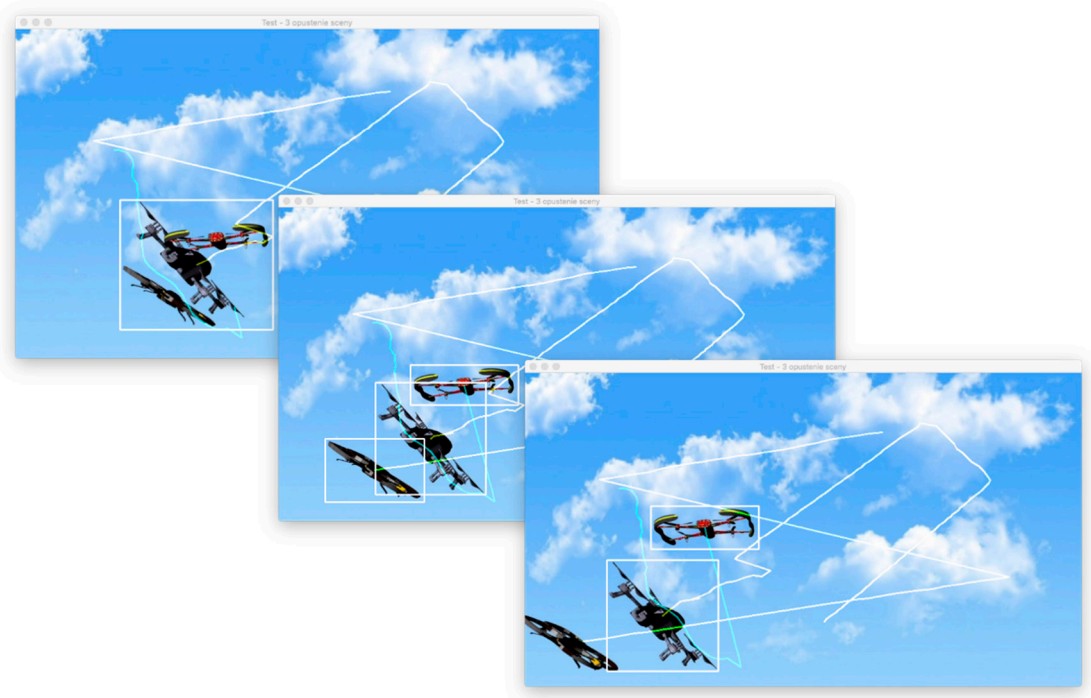

**Figure 11.** Overlapping of objects.

In Figure 11 (first picture) there was an overlap of objects and the proposed detection procedure evaluated all objects as one. In this step, the object that had the largest match in the previous frame was identified. In this case, it was the drone in the center of the group. Because there was only one object identified in the scene, the others were not tracked. In the next frame, when objects were separated, some objects were re-matched with objects in the database and some of them were evaluated as new ones.

After verifying the detection procedure in various scenarios, the reliability of the selected ORB detector and its BRIEF descriptor was further evaluated. The goal was to evaluate the number of correctly matched features throughout the object's occurrence on the scene. In this way, the scenario where the object came out of the sensing area several times and returned in a different angle, and in a different size, was evaluated. The maximum number of features detected and recorded in the database was set to 500. The number of correctly matched features will, therefore, be in the range of 0–500. The test video is 16 s long and consists of 481 frames. During the whole test (Figure 12), 500 features were detected for a single object in each frame and of course the same number was recorded in the database.

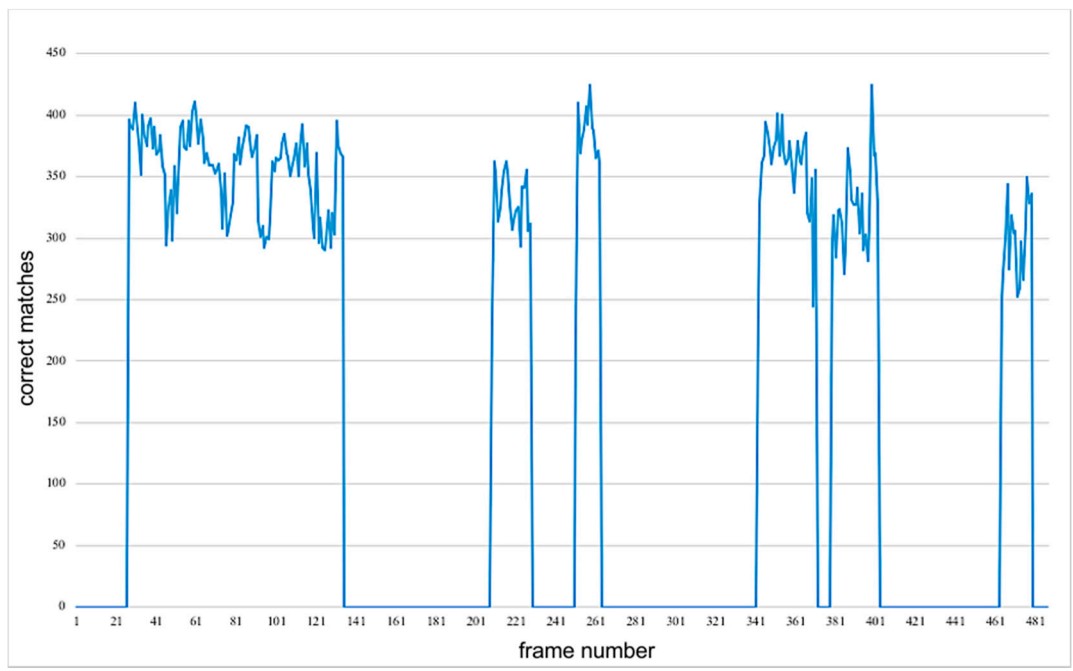

**Figure 12.** The number of correctly matched features in 481 frames.

The results (Figure 12) are divided into 6 sections during which the object was in the sensing area. During the presentation in the first area, the object was stable, partially enlarged and reduced in size, and did not significantly change its position. This is also reflected in the relatively stable number of correct matches found. During the object's presence in Sections 2–5, it moved away from the sensing area for only a short time and it always returned relatively with the same rotation. While in the last area, the object rotated all the time, changing its size and position. For this reason, the number of correct matches found is the smallest (in the range of 250—350). This once again shows that the selected detector is effective even in situations where an object changes its size, position, or rotation during tracking.

The proposed solution relies on the usage of classical computer vision methods, which are not intended exclusively for object tracking. For this reason, there were several situations during validation where the implemented tracking procedure did not behave as was required. Specifically, this was the case in a situation where objects clashed and, after separation, they were incorrectly paired with objects in the database. Therefore, it would be appropriate to use a method where it is possible to identify the object in the scene clearly and thus make it easier to track. It would also be appropriate to use a method other than background subtraction to identify a moving object in the image because of the susceptibility of this method to scene changes, such as lighting conditions or various camera noise when focusing, and so on. Therefore, other detection and identification principle was proposed and tested.

Used methods for drone tracking and detection methods were mainly tested on the simulated scenes described above. This work aims to use and verify the success of detection multi-rotor UAVs using machine-learning methods, so in the next section, we will describe the process of training and evaluation on a real dataset of photos and videos.

## 4. Detection and Identification of a Drone Using a Machine-Learning Approach

The goal of using machine-learning methods is to choose a suitable neural network, which is acceptable for the detection of multi-rotor UAV machines. For this purpose, we have chosen an available and modern TensorFlow machine learning platform. After selecting a suitable type of network, our task was to overtrain it using appropriate network parameters and test its success ratio in

classifying objects moving in the sky into the considered classes. Our solution and proposed approach consist of the following steps:

- preparing data for training;
- preparing data for evaluation;
- selection of detection model;
- creating other necessary files for training;
- training;
- export of the trained model to a frozen graph format;
- creating an application to test the detector.

### 4.1. Preparing Data for Training and Evaluation

For successful training of a neural network, a sufficiently large set of data is needed. The object of interest should be captured in various light, spatial or other conditions. It is also important that the data show all of the defined drone types in different situations, rotations, and environments. The amount of data collected should be sufficient to allow the network to be trained and to recognize the object of interest. In addition to the obtained data, it is also necessary to make annotations that specify the exact location of the object in the scene.

### 4.2. Selection of Detection Model

Since creating a brand-new recognition model would require a large amount of time and computing power to achieve the desired results, detection models that were trained on the data set called "Common Objects in Context dataset" were used after analyzing the available solutions. This set contains over 200,000 annotated images of various classes (e.g., cat, dog, car, boat, etc.) that can be used for the training. However, the drone is not among them. On the other hand, defining our detection model would probably be more reliable than using the pre-trained model, as all the training parameters would be adopted to specific detection.

In Table 2 some of the latest available detection models are listed with the time that represents the speed of a particular model for a sample of 600 × 600-pixel images and also bounding box mean average precision (mAP) representing the performance of the detector. The speed measurement was performed with the NVidia GeForce GTX TITAN X graphics card and mAP metric evaluated on COCO 14 minimal set. The table contents are taken from the official TensorFlow repository on the Github software development platform.

**Table 2.** Pre-trained TensorFlow models.

| Name of the Model | Speed of the Model [ms] | Mean Average Precision [1] |
|---|---|---|
| ssd_mobilenet_v2_coco | 31 | 22 |
| ssd_inception_v2_coco | 42 | 24 |
| faster_rcnn_inception_v2_coco | 58 | 28 |
| faster_rcnn_resnet50_coco | 89 | 30 |
| faster_rcnn_resnet50_lowproposals_coco | 64 | - |

[1] See evaluation protocol here: https://cocodataset.org/#detection-eval.

Each of these models contains a configuration file that serves as a source of information for training the model. When selecting the parameters of the model, it is necessary to consider the specific application. Models such as Faster Regions with Convolutional Neural Networks (R-CNN) use the selective search method to find possible objects in a scene and they are used when the detector is required to be more accurate. On the other hand, the processing time in these models has lower priority. This is the case of automated UAV detection, so a model of this type will be used. By contrast, SSD (Single Shot MutliBox Detector) models have priority on the processing time. This is proven by the

table of pre-trained models (Table 2, where the abbreviation SSD or RCNN is used in the model name and this corresponds to the model type) [21].

### 4.3. Creating Other Necessary Files for Training

For successful training of the model, it is necessary to have a sufficient amount of data. Therefore, drone images were collected from freely available sources on the Internet. Their number was approximately 100. After communication with the developers of the SafeShore project, a project funded by the European Commission to detect small targets flying at low altitudes, a further data set was obtained of approximately 1000 samples. Other samples were created from video sequences capturing a flying drone.

Next, it was necessary to create our annotation for each image showing the object of interest—the drone. The data was split so that 80% of the data was left for model training and the rest 20% for testing. The freely available LabelImg tool was used to create annotations. An example of an annotation can be seen in Figure 13.

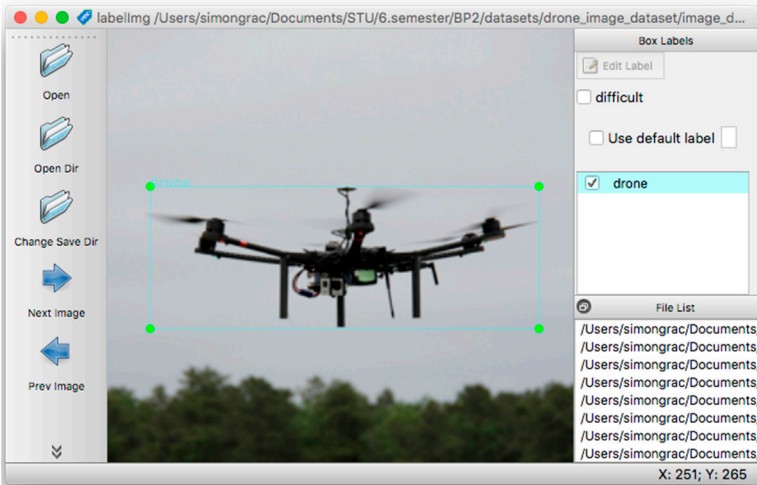

**Figure 13.** Annotated object (drone) in LabelImg.

The .xml file generated by the annotation tool can be seen in Figure 14. From this file, the data, which are used to create the TensorFlow Record (TFRecord) file, were extracted. It is the simple file whose content is in binary form and it is used by the TensorFlow training library. Therefore, it is located in one memory block and the data access is faster. The extracted data are saved in a .csv file (right part of Figure 14). This file displays information such as the name of the file in which the drone object is located, its height and width, the name of the class, and the coordinates of the upper right corner of the located object.

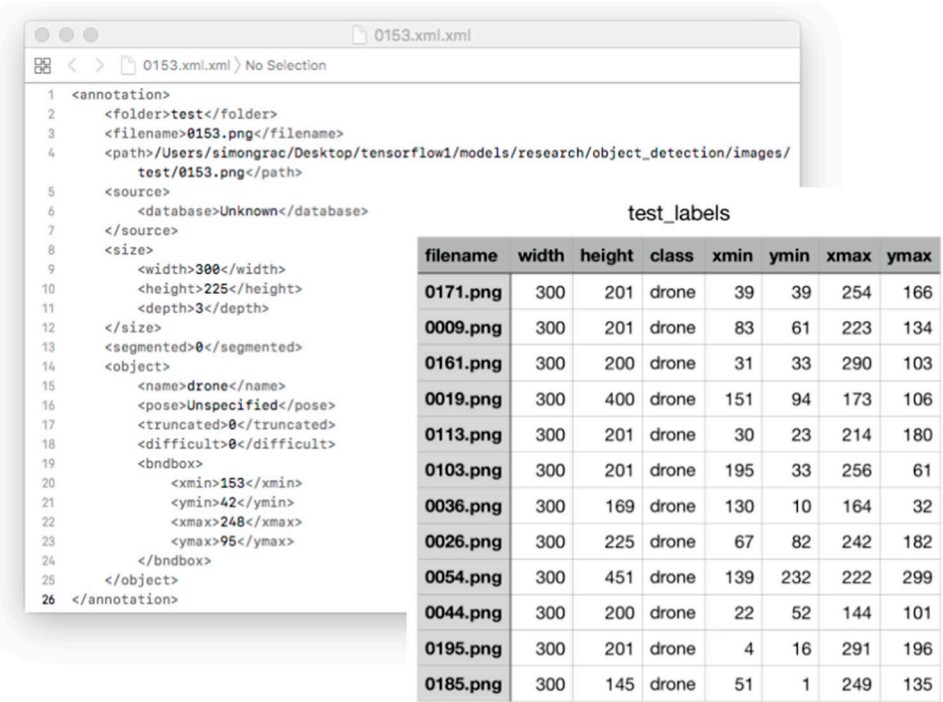

**Figure 14.** The creation of TensorFlow Records (TFRecords).

## 4.4. Training

The training script is provided directly by the library developers and it is included in the official Github repository. The time required to complete one training step took 4.5 s on average (using TensorFlow with a Central Processing Unit (CPU) support only—3,5 GHz Intel Core i5). Using TensorFlow with a Graphics Processing Unit (GPU) support—NVIDIA GeForce GTX1050 Ti, the time of one training step was on average 0.33 s, so we use this approach. The difference between the training time is, therefore, highly dependent on how the detector is trained. Information about model and training configuration is summarized in Table 3. For more information about model architecture, see publication [23] from Szegedy et al.

**Table 3.** Model and training configuration.

| Model Configuration | |
|---|---|
| model name | Faster R-CNN with Inception v2 |
| 1st stage (location) regularizer | L2 |
| 1st stage initializer | Truncated Normal |
| 2nd stage (classification) regularizer | L2 |
| 2nd stage initializer | Variance Scaling |
| score converter | SOFTMAX |
| **Training Configuration** | |
| learning rate | 0.0002 |
| number steps | 300k |

A model with adequate reliability and feasibility was achieved after a total of 300,000 steps (about 2 training days). The progress of classification and object location is shown in Figure 15. The data on the graph entitled Classification loss show a gradual improvement of the model according to the number of training steps in the task of classifying the object into the correct class. In the case of a model that classifies data into a single class (single class classification), objects that do not contain the drone

are considered as incorrectly classified. Another graph called Localization loss shows how the model was improved with the number of training steps in the task of a correctly localized object.

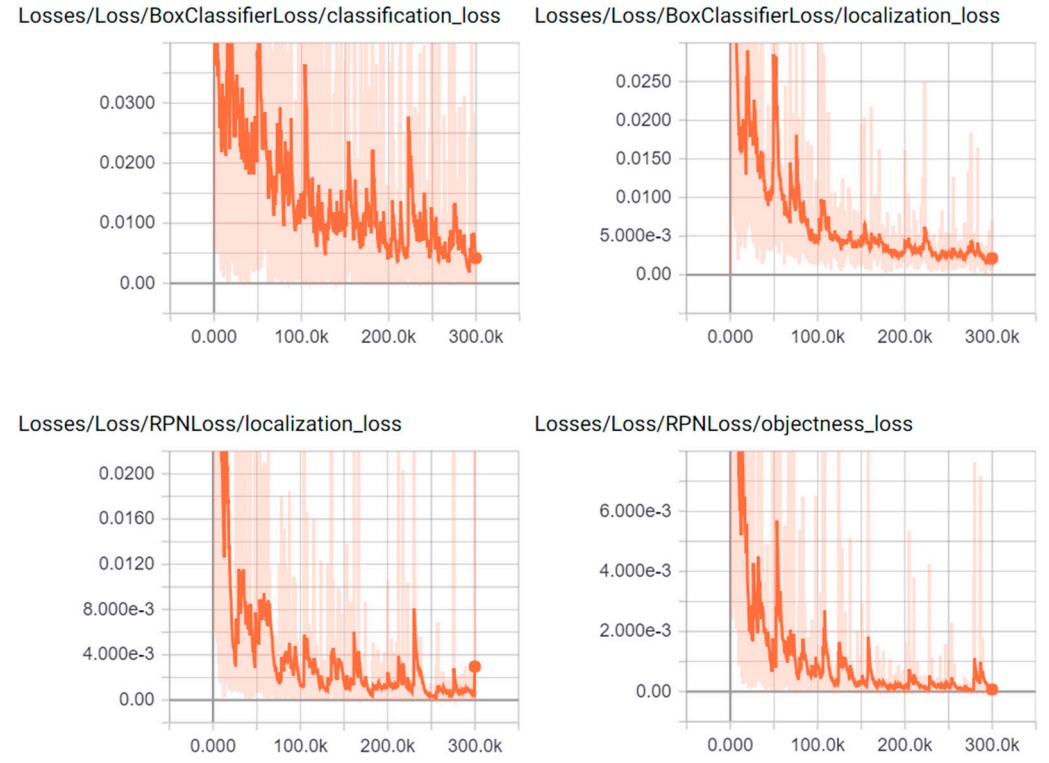

**Figure 15.** Progress of the training in drone detection using TensorFlow.

The graph in Figure 16 shows the improvement of the detection model taking into account all aspects of the success and correct detection of the object.

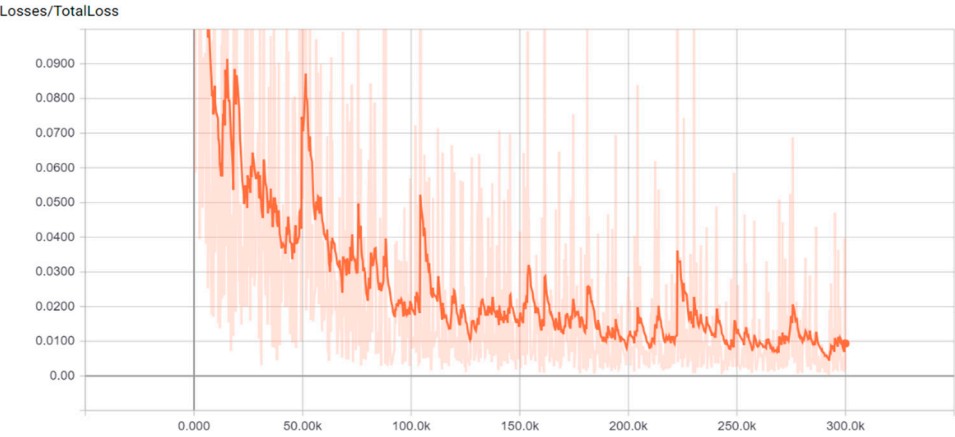

**Figure 16.** Progress of the model's training concerning the overall aspects of reliable and accurate object detection.

## 5. Experiments

Various scenarios were verified to determine the success rate of the trained detection model under different conditions. The model was tested on drone images and videos as well as various backgrounds. The testing scenarios were as follows:

- drone in a static image;

- drone in the sky;
- multiple drones in one image;
- drone with another flying object in the sky.

## 5.1. Drone in a Static Image

Static images of a drone with varying background and several other objects on the scene were presented to the detector. In most cases, the drone was detected correctly. The successful finding of the object in several pictures is shown in Figure 17.

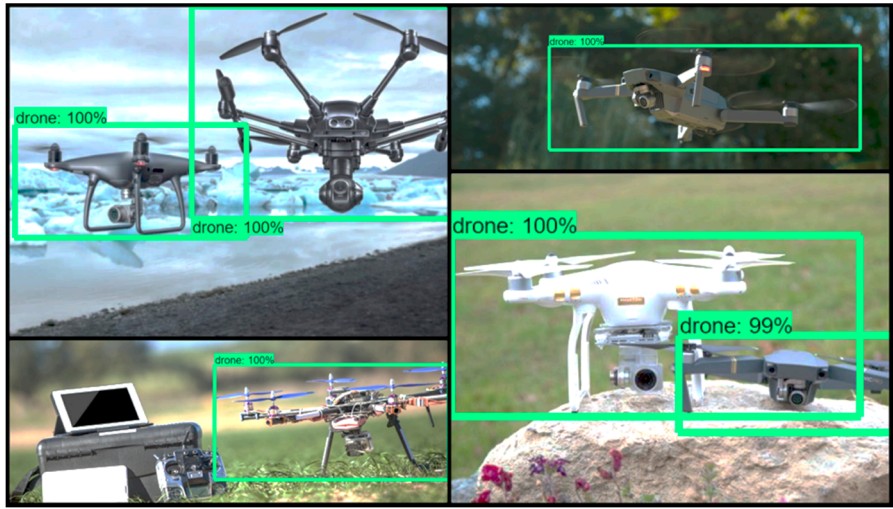

**Figure 17.** Detection of the drone in the static image.

During testing, there were also situations when the drones were detected incorrectly. Specifically, such situations occur where one drone was split into two objects or a boat object was identified as a drone (Figure 18).

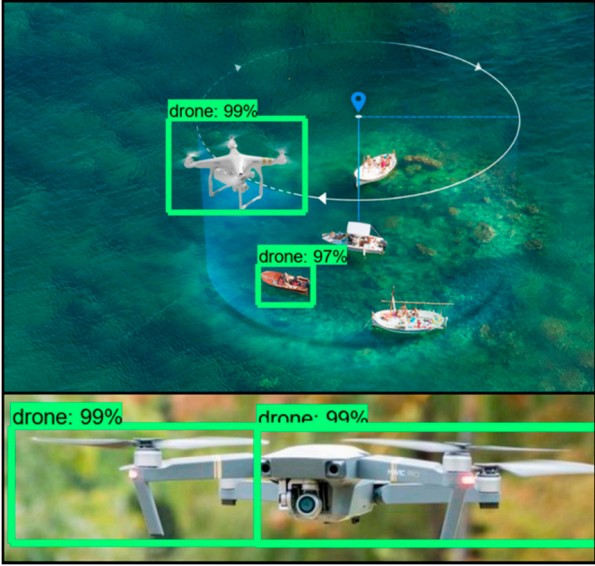

**Figure 18.** Incorrect drone detection in a static image.

### 5.2. Video: Drone in the Sky

This scenario was focused on the classic situation when the camera is pointing to the sky and the drone is moving freely in the sensing area. During the whole scenario, there was no other object in the image. A video from a series of data from the SafeShore project developers was used for testing. The drone was detected correctly during the whole scene. One of the detection results is shown in Figure 19.

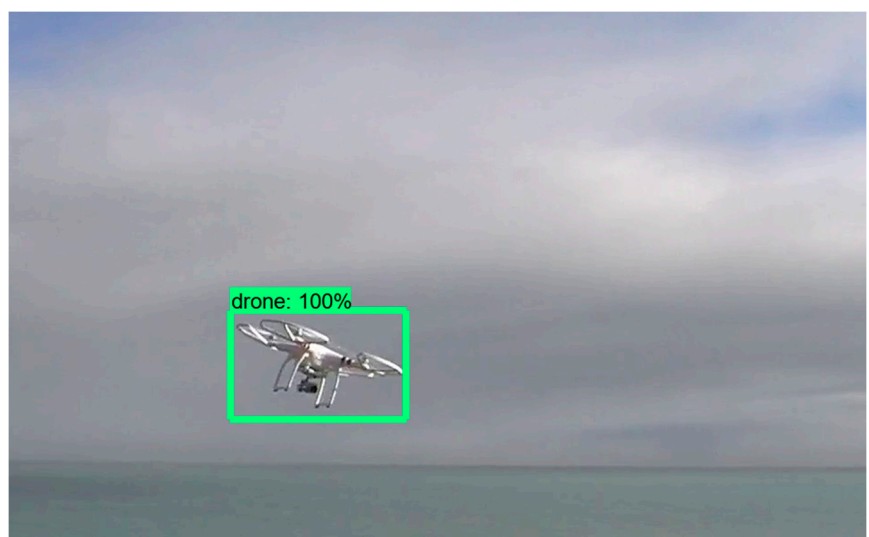

**Figure 19.** Detection of the drone in the sky.

In this scenario, we also monitored the success of our detection model depending on the size of the drone in the scene. In this case, the size of the drone corresponded to its distance from the camera sensor. During this test, the object occupied from 0.42% to 3.43% of the image area. Detector operated reliably at every object size. Examples of drone sizes in the test video are shown in Figure 20.

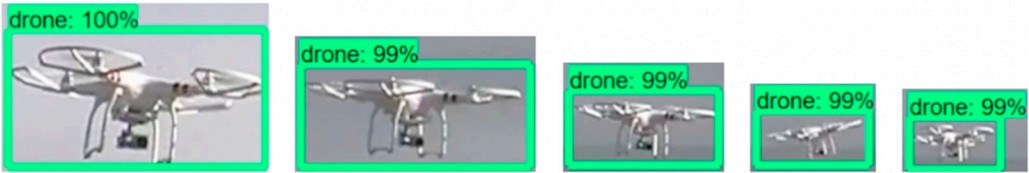

**Figure 20.** Drone sizes.

### 5.3. Video: Multiple Drones in One Image

When testing this scenario, multiple drones occurred in the sensing area. All the drones were correctly detected and bounded during the whole testing process. A screenshot from the testing is shown in Figure 21.

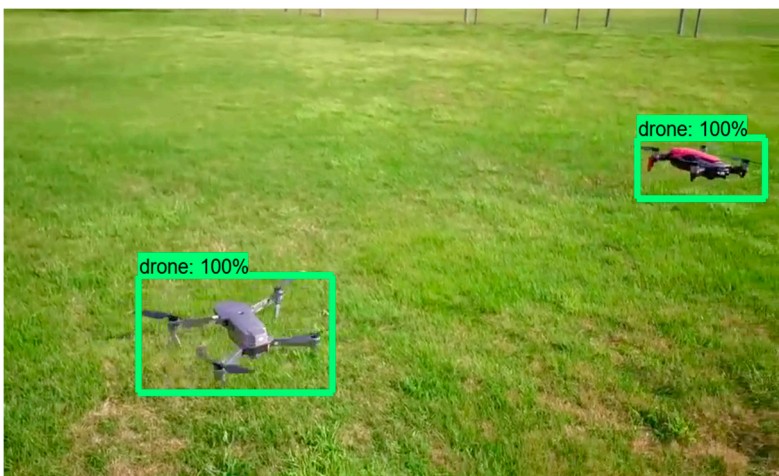

**Figure 21.** Testing of multiple drones in the scene scenario.

*5.4. Video: Drone with Another Flying Object in the Sky*

In this scenario, it was investigated how the detector reacts to a situation when several objects of a different kind appear in the sensing area. The detector was tested on different objects and different classes. A situation where the drone and the bird appeared on the scene at the same time was also tested. These two objects are each of a different class, but visually they are very similar. Therefore, the proposed detector failed in this situation. As shown in Figure 22, the detector classified the bird in the same class as the drone.

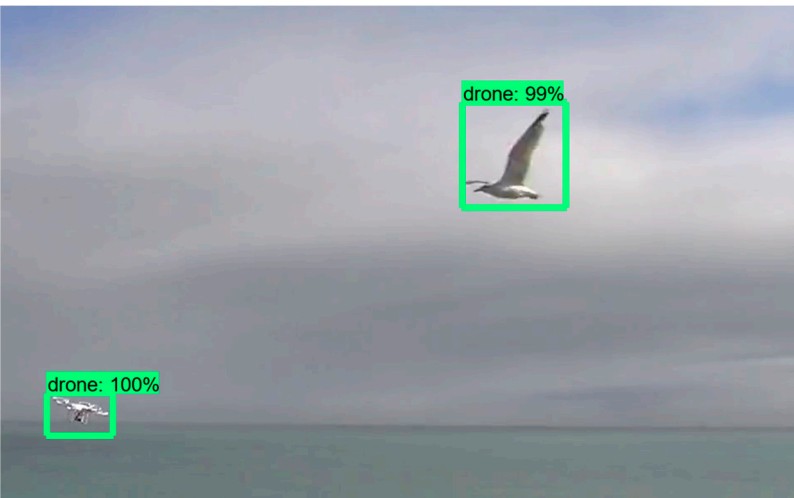

**Figure 22.** Testing the drone detection in the sky with another object.

For this reason, one more detector has been trained to be able to recognize two classes of objects, namely the drone and the bird. After training, the test was repeated. As shown in Figure 23, the detector was successful in this case and it classified the objects into the correct classes.

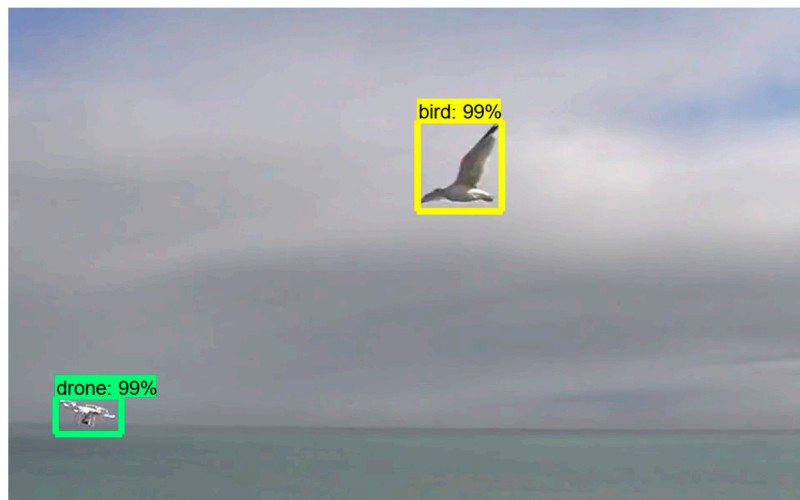

**Figure 23.** Testing the objects detection in the sky with an improved detector.

### 5.5. Statistical Evaluation of the Detector

The detection rate of two detectors was statistically evaluated—a detector able to classify objects into one class (drone) and a detector able to classify objects into two classes (drone and bird). The detectors were evaluated on a set of 150 images; 74 of them represented a single drone, and 76 represented a bird. For this evaluation, a new data set was used that was not used in training. The detection rate of the detector trained to one class is shown in Table 4 and the same rate of the detector trained to two classes is shown in Table 5.

**Table 4.** The detection rate of the detector trained to one class (drone).

| Object Class | Num. of Objects | Num. of Successful Detections | Num. of Failed Detections | Detection Rate [%] |
|---|---|---|---|---|
| Drone | 74 | 74 | 0 | 100.0 |
| Bird | 76 | 18 | 58 | 23.6 |
| Overall | 150 | 92 | 58 | 61.3 |

**Table 5.** The detection rate of the detector trained to two classes (drone and bird).

| Object Class | Num. of Objects | Num. of Successful Detections | Num. of Failed Detections | Detection Rate [%] |
|---|---|---|---|---|
| Drone | 74 | 70 | 4 | 94.5 |
| Bird | 76 | 76 | 0 | 100.0 |
| Overall | 150 | 146 | 4 | 97.3 |

SafeShore licensed data were used for testing as well as training. The first set of test images were displaying one or more instances of a multi-rotor drone object. There were also other objects in the pictures, such as people, water, grass, nature, other machines, etc. An example of this data is shown in Figure 24. Some drones in testing pictures were also equipped with various add-ons. The second set of pictures contained birds of various sizes, often with sky as the background.

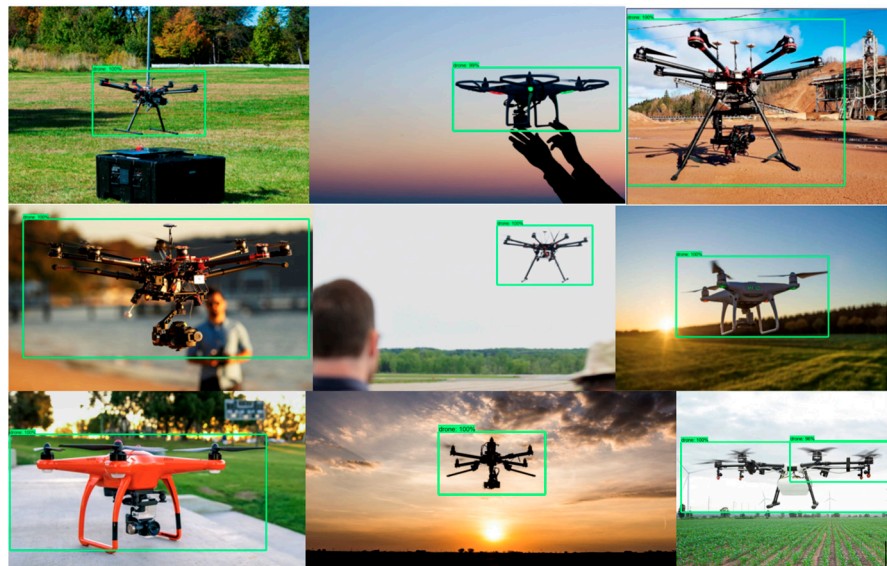

**Figure 24.** Testing the objects detection in the sky with an improved detector.

## 6. Conclusions and Future Work

A reliable detection mechanism of multi-rotor UAV was achieved by the usage of the TensorFlow library. For this purpose, the "Common Objects in Context dataset" was used and it was extended by 1000 samples of UAVs from the SafeShore dataset. A successful detection rate of 97.3% in optimal conditions was achieved.

It can be concluded that the proposed detection principle was successful in described testing cases. The trained detection model was successful in almost every tested scenario. A problem occurred only when objects with similar features from another class were on the scene. Statistical evaluation of the detection model showed that the detection rate, in this case, was only 61.3%. Moreover, the bird should not be detected at all. Therefore, another detection model was used. This model was able to recognize two classes—drone and bird. From the experiments, it was found that if the detector was trained on several objects, the detection was much more successful. From this finding, we conclude that the biggest problem in drone detection is the occurrence of objects with similar properties. These objects comparable to the drone can include a bird, airplane, parachute, or paragliding wing. There are not so many of these objects, so it would be possible to train such a detection model with a sufficiently large data set. Such a detector can be even more successful than the one trained in two classes. On the other hand, the detection rate of 97.3% of the detector trained in two classes is sufficient for most of the basic security applications. Achieved success and system reliability can be compared with the common human observer. Testing was performed on all described data and conditions and our work presents a functional approach for multi-rotor drone detection under ideal flight conditions. Future work using our approach may be to focus on the cases described in the chapter Introduction, where the detector could represent not only the role of a common observer but also a sophisticated detection system usable in adverse conditions.

**Author Contributions:** Šimon Grác was the main investigator. Peter Beňo mainly proposed the methodology used in the article. František Duchoň supervised the article, Martin Dekan provided the validation of the results and Michal Tölgyessy was responsible for the text reviewing and editing. All authors have read and agreed to the published version of the manuscript.

**Funding:** This work was supported by projects APVV-17-0116, VEGA 1/0752/17, and DIH$^2$.

**Conflicts of Interest:** The authors declare no conflict of interest.

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
