# Peer review of "Automated Detection of Multi-Rotor UAVs Using a Machine-Learning Approach"

_asi, doi:10.3390/asi3030029_

Round 1

Reviewer 1 Report

The paper is in general well written and technically sound. However, there exist a number of minor problems hindering the quality. For example, the authors should give more details about the results of the model testing. The paper has mentioned that training methods including SSD and Faster RCNN, which belong to object detection, and show the speed of the different model, but didn't give mAP(mean Average Precision).

Author Response

Dear reviewer,

We appreciate your feedback and valuable comments. The requested mAP metric for described training models has been added to the article.

Despite the fact that it would be possible to give even more details about tested models than we provided, we have mentioned the most important ones according to the application’s main use case. We would be able to use another metric for testing our model, IoU for example, but due to short response time to the opponents' comments, we do not plan to realize it for now.

A sufficient number of experiments were performed to verify the success of our detection model. Firstly, with graphic simulations that display drones in various situations and later in real-world conditions as real images and available videos displaying drones in various situations/sizes/appearance states, etc. The system should primarily detect the presence/absence of the drone in the image (or video). For our application, it is not very important to verify the full accuracy of the detected position of bounding boxes, as it is important in face recognition applications for example. The priority is to differentiate the drone from objects of a similar kind and correctly evaluate its presence.

We believe we have explained it enough. Thank you again.

Reviewer 2 Report

The work could be improved by adding the result of testing networks with different learning parameters. For example, it would be interesting to include the mathematical definition and values obtained from Classification Loss in a table together with the Learning Rate of the network.
An example is given in the following reference using CNN:

  Volume 2019 |Article ID 7206096 | 11 pages https://doi.org/10.1155/2019/7206096

Secure UAV-Based System to Detect Small Boats Using Neural Networks

Author Response

Dear reviewer,

We appreciate your feedback and valuable comment. Such experiments have been performed, but due to the limited paper scope, it is not possible or even purposeful to display all of them in the paper (specifically, the most experiments were performed with the learning rate parameter). Only the optimal achieved results are presented. Our main goal was to prove the effective application of the selected detection model. We hope that this answer and explanation will satisfy your demand for more results.

Thank you again.

Round 2

Reviewer 2 Report

In order to adequately support the conclusions provided, it is necessary to include in the paper some information about the training and the type of neural network(s) used.
As the authors themselves respond:
"Such experiments have been performed, but due to the limited paper scope, it is not possible or even purposeful to display all of them in the paper (specifically, the most experiments were performed with the learning rate parameter). Only the optimal achieved results are presented."
Therefore, it would not be very difficult to provide a table with such results, justifying the optimal results, improving the final work.

Author Response

Dear reviewer,

The requested data about the neural network was added to the article. It is highlighted in green.

Round 3

Reviewer 2 Report

Ok

Author Response

The article was modified according to the Academic Editor Notes.

This manuscript is a resubmission of an earlier submission. The following is a list of the peer review reports and author responses from that submission.

Round 1

Reviewer 1 Report

Good for the publication in the present form

Reviewer 2 Report

Well done.

Reviewer 3 Report

The paper presents an attempt to automate visual detection of drones in the sky through automated image analysis. The main objective of this work is to develop a solution against intruding drones and enable protection of infrastructure and privacy. While this is a good goal, the paper contains multiple flaws in both their analysis the problem, which is oversimplified, and, as a result, in their development of a solution that would only work on a very limited scale. Furthermore, at this stage the manuscript suffers from poor referencing and requires editing for grammar and style. As a result, I recommend that the paper be rejected at this point, with a recommendation to the research team to focus on more realistic UAV intruder scenarios and improve the technical quality of the manuscript.

The broad issues are outlined below:

  1. It is a fundamentally flawed assumption that UAVs are all multirotor systems with vertical take-off and landing capabilities. Today, a considerable number of surveillance drones with high resolution imaging capabilities are fixed winged drones (that may have VTOL capabilities). This UAV category is ignored by the research team.
  2. The first issue should have been addressed in the Introduction section of the manuscript, where previous studies and available UAV system types should be mentioned. At this stage, there no references (!!) in the Introduction section of the manuscript. In fact, there are only 12 references provided in the manuscript at this stage
  3. There is a clear misunderstanding on the part of the authors as to how actual intruder UAVs behave in a malicious operations scenario. Firstly, drones may be flying low to the ground, obscured by terrain or anthropogenic obstacles. Or, on the contrary, UAVs may approach the target at a high altitude, where they could be obscured by cloud or smoke screen cover. Furthermore, intruder drones can approach simultaneously from multiple directions at variable altitudes, and steps may be taken to add visual camouflage elements to their structures Finally, and perhaps most obviously, intruder UAVs can be deployed at night and equipped infrared cameras for detailed surveillance of structures, vehicles, and people – this basic scenario is not discussed in the paper at all.

Overall, while the presented type of system of visual could potentially warn of accidentally approaching hobbyist drones during daytime hours with unobscured line of sight, its effectiveness against UAVs launched with specific malicious intent and appropriate strategies and counter-measures would likely be limited. Without testing these scenarios, the study is of very limited practical value.

Reviewer 4 Report

The publication lacks a literature review. Discussion of available methods and selection of the method used requires justification.
The experiment should be better described. Please complete the conclusions.

Reviewer 5 Report

  1. The title of the article is too broad. You should try to put more keywords that reflect on the main methods you are using or make it more specific. Otherwise, it looks like a review paper.
  2. I think the distance of UAV to the camera or the size of UAV in the image will significantly influence the result of the detection. Have you studied the effect of distance? What is the maximum distance or size of UAV in the image could you make a relatively high detection?
  3. Are there any restrictions for the 97.3% successful rate mentioned in the abstract? Will this number apply to all conditions such as distance, light, or number of UAVs in the image?
  4. Could you make more explanation about the "unsolvable situations" in the abstract rather than put this general reason.
  5. How did you get the result in Table 1? It looks like the first two methods 'SIFT' and 'SURF' do not have any advantages.
  6. The title of Section 4 is also too broad. ‘AI’ is a very large term and it could include many aspect. I don't think using TensorFlow to do anything could be called 'using AI'. 
  7. From Fig 8-11 you used the simulated pictures, but Fig 13 used the real photos, what is the reason for the different image sources? Could you also use the real photos for the computer vision methods?
  8. I think Fig 15 is not necessary and important.
  9. It is possible to open source the images you used to train the model and used to verify the result in Table 3?
  10. Could you get some conclusion about under what kind of scenarios are the UAV easy to be detected and what reasons caused the fail in Table 3? If there are other objects other than birds such as buildings and trees, will they influence the detection rate?